# FGF-2 Differentially Regulates Lens Epithelial Cell Behaviour during TGF-β-Induced EMT

**DOI:** 10.3390/cells12060827

**Published:** 2023-03-07

**Authors:** Mary Flokis, Frank J. Lovicu

**Affiliations:** 1Molecular and Cellular Biomedicine, School of Medical Sciences, Faculty of Medicine and Health, The University of Sydney, Sydney, NSW 2006, Australia; 2Save Sight Institute, The University of Sydney, Sydney, NSW 2006, Australia

**Keywords:** transforming growth factor-beta (TGF-β), fibroblast growth factor (FGF), epithelial-mesenchymal transition (EMT), fibrosis, cataract, lens

## Abstract

Fibroblast growth factor (FGF) and transforming growth factor-beta (TGF-β) can regulate and/or dysregulate lens epithelial cell (LEC) behaviour, including proliferation, fibre differentiation, and epithelial–mesenchymal transition (EMT). Earlier studies have investigated the crosstalk between FGF and TGF-β in dictating lens cell fate, that appears to be dose dependent. Here, we tested the hypothesis that a fibre-differentiating dose of FGF differentially regulates the behaviour of lens epithelial cells undergoing TGF-β-induced EMT. Postnatal 21-day-old rat lens epithelial explants were treated with a fibre-differentiating dose of FGF-2 (200 ng/mL) and/or TGF-β2 (50 pg/mL) over a 7-day culture period. We compared central LECs (CLECs) and peripheral LECs (PLECs) using immunolabelling for changes in markers for EMT (α-SMA), lens fibre differentiation (β-crystallin), epithelial cell adhesion (β-catenin), and the cytoskeleton (alpha-tropomyosin), as well as Smad2/3- and MAPK/ERK1/2-signalling. Lens epithelial explants cotreated with FGF-2 and TGF-β2 exhibited a differential response, with CLECs undergoing EMT while PLECs favoured more of a lens fibre differentiation response, compared to the TGF-β-only-treated explants where all cells in the explants underwent EMT. The CLECs cotreated with FGF and TGF-β immunolabelled for α-SMA, with minimal β-crystallin, whereas the PLECs demonstrated strong β-crystallin reactivity and little α-SMA. Interestingly, compared to the TGF-β-only-treated explants, α-SMA was significantly decreased in the CLECs cotreated with FGF/TGF-β. Smad-dependent and independent signalling was increased in the FGF-2/TGF-β2 co-treated CLECs, that had a heightened number of cells with nuclear localisation of Smad2/3 compared to the PLECs, that in contrast had more pronounced ERK1/2-signalling over Smad2/3 activation. The current study has confirmed that FGF-2 is influential in differentially regulating the behaviour of LECs during TGF-β-induced EMT, leading to a heterogenous cell population, typical of that observed in the development of post-surgical, posterior capsular opacification (PCO). This highlights the cooperative relationship between FGF and TGF-β leading to lens pathology, providing a different perspective when considering preventative measures for controlling PCO.

## 1. Introduction

The ocular lens is a transparent, avascular tissue responsible for transmitting light onto the retina. It contains two cell types: cuboidal epithelial cells and adjacent elongate fibre cells, both comprised of specialized molecular (e.g., crystallins) and cytoskeletal (e.g., intermediate filaments) properties to facilitate vision [1]. Ocular growth factors, such as fibroblast growth factor (FGF) and transforming growth factor-beta (TGF-β), are key regulators of different cellular processes in the lens, including epithelial cell proliferation [2,3,4], fibre differentiation [1,5,6,7,8,9,10], and epithelial–mesenchymal transition (EMT) that lead to lens pathology [11,12,13,14,15,16,17]. In situ, FGF is thought to be required for regulating normal lens cell processes in a spatially dependent manner, as previously reviewed [1].

TGF-β can regulate and/or concurrently dysregulate normal lens homeostasis, cell growth, and survival, by altering lens epithelial cell (LEC) morphology [17,18,19]. The dysregulation of lens epithelial cell architecture induced by TGF-β is characterized by EMT, a phenomenon that has been widely reviewed [20,21,22], with normal cuboidal LECs transitioning to become aberrant migratory, contractile myofibroblastic cells. These myofibroblastic cells can aggregate to form a fibrotic plaque leading to cataracts [23,24,25]. To date, cataracts, that have been extensively reviewed and studied, are still considered the most common form of blindness worldwide [26,27,28], with the only form of treatment being surgical intervention. Despite the effectiveness of surgery, posterior capsular opacification (PCO), known also as a secondary cataract, may result post surgery, requiring further intervention [29,30,31]. PCO results from the aberrant behaviour of LECs left after surgery, with these cells either undergoing EMT to form a posterior subcapsular plaque (fibrotic PCO) [11,24,25,32], or differentiating into aberrant fibre cells leading to Elschnig’s pearls and Soemmerring’s ring (regenerative, pearl PCO) [33], as previously reviewed [34,35]. While these two different spatially distinct epithelial PCO pathologies are well characterised [36,37], the underlying molecular mechanisms regarding their formation are poorly understood.

Numerous models have established TGF-β-induced lens EMT responses in humans [14], embryonic chicks [13,38], and murine cell lines and explant models [13,15,25,32,39]. In dissociated embryonic chick lens epithelial cells treated with TGF-β, we see a heterogenous response, with some cells undergoing fibre differentiation, while others undergo EMT [13,40]. In in vitro studies using mammalian lens epithelia, exogenous treatment of LECs with TGF-β results in a homogenous EMT response [17,41,42,43]. In situ, however, anterior subcapsular cataracts (ASCs) develop in transgenic mice in response to elevated activity of ocular TGF-β [44]; the subcapsular plaques are comprised a heterogenous population of aberrant lens fibre cells and myofibroblastic cells, similar to those seen in human cataracts [24]. The in situ transgenic mouse model ideally replicates the human clinical pathology of fibrotic cataracts, that is attributed to the endogenous ocular milieu of different growth factors and cytokines, that do not act in isolation, unlike what we have in vitro. Since two disparate lens epithelial phenotypes contribute to ASC and PCO, it is important to better understand how they are derived, and the putative interplay of the different ocular factors involved.

While FGF is well established in regulating lens epithelial cell proliferation and fibre differentiation, it has also previously been shown to influence TGF-β-induced EMT and aberrant cell behaviour, promoting wound healing, repair, and fibrogenesis [5,16,41,45]. For example, different relatively low doses of FGF-2 (2.5–20 ng/mL) can exacerbate TGF-β2 (0.5–3 ng/mL)-induced lens opacification in intact rat lenses, with the higher dose combinations exhibiting the most pronounced response, resulting in dense cellular plaques and elevated deposition of ECM [16]. In contrast, other studies have shown that FGF can counteract and antagonise EMT in rodent LECs [14,15]. Rat lens epithelial cell explants cotreated with a relatively low dose of TGF-β2 (50 pg/mL) and a lower dose of FGF-2 (10 ng/mL) still formed spindle-like cells typical of EMT; however, with minimal cell loss compared to explants treated with TGF-β2 alone [15]. This increased cell survival was unique to FGF as other regulatory ocular growth factors (e.g., EGF, IGF, HGF, or PDGF) could not block the hallmark features of TGF-β-induced EMT, including lens capsular wrinkling, apoptosis, and cell loss [15,46].

The influence of FGF regulating TGF-β-induced EMT may be attributed to the putative crosstalk between various downstream intracellular signalling pathways; the TGF-β-canonical Smad2/3-dependent proteins, and non-canonical mitogen-activated protein kinases (MAPK), such as extracellular signal-regulated kinase (ERK1/2) [14,38,47,48,49]. Studies using mouse LEC lines (MLECs) showed that cotreatment of cells with FGF-2 (10 ng/mL) and TGF-β2 (10 ng/mL) resulted in elongated fibroblastic-like cells and enhanced cell migratory mechanisms, with elevated ERK1/2-signalling [14]. Interestingly, in human lens epithelial cells (HLECs) from this same study, cotreated with the same doses of FGF-2/TGF-β2, they report on the antagonistic behaviour of FGF-2 with a reduction in cytoskeletal markers involved in stress fibre formation [14]. It is clear from these studies that there is no consistency in cell responsiveness to both FGF/TGF-β across different species.

In the current study, we characterized the influence that FGF has on TGF-β-induced cell behaviour in rat lens explants to best model the conditions needed to promote a heterogenous cell population typical of fibrotic cataracts as seen in situ. We demonstrate that a high fibre-differentiating dose of FGF is protective of TGF-β-induced EMT in peripheral lens epithelia; however, this is not evident in central lens epithelia induced by TGF-β. This emulates the spatial phenotypic response of lens cells seen in human PCO and may serve as a model to better understand the mechanisms leading to this post-surgical pathology.

## 2. Materials and Methods

### 2.1. Animals and Tissue Culture

All procedures conducted abided by the Australian Code for animal care and usage for scientific purposes and the Association for Research in Vision and Ophthalmology (ARVO) Statement for the Use of Animals for Ophthalmic and Vision biomedical research (USA). The experiments were approved by the Animal Ethics Committee of The University of Sydney, NSW, Australia (AEC# 2021/1913). Wistar rats (*rattus norvegicus*) at 21-days of age (P21 ± 1 day) were humanely euthanized with CO_2_ followed by cervical dislocation.

### 2.2. Lens Epithelial Explants

All collected primary rat ocular tissues were kept in medium 199 with Earle’s Salts (M199) (11825015, Gibco^TM^, Thermo Fisher Scientific, Sydney, NSW, Australia) in 35 mm Nunc™ culture dishes (NUN150460, Thermo Fisher Scientific). The media was supplemented with 2.5 μg/mL Amphotericin B (15290-018, Gibco^TM^, Thermo Fisher Scientific), 0.1% bovine serum albumin (BSA) (9048-46-8, Sigma-Aldrich Corp., St. Louis, MO, USA), 0.1 μg/mL L-glutamine (200 mM) (25030081, Gibco^TM^, Life Technologies, Carlsbad, CA, USA), and penicillin (100 IU/mL)/streptomycin (100 μg/mL) (15140-122, Gibco^TM^, Life Technologies). The collected eyes were placed under a dissecting microscope to remove the lenses. The posterior capsule of the lens was torn using fine forceps and the remaining intact anterior capsule containing a sheet of lens epithelial cells (LECs) was pinned to the base of the culture dish using the gentle pressure of the forceps, as previously described [5]. Explants were maintained in a humidified incubator (37 °C, 5% CO_2_).

Different doses of recombinant human TGF-β2 (302-B2-002, R&D systems, Minneapolis, MN, USA) were used to induce EMT in the lens explants, as previously described [39]. A lower dose of TGF-β2 (50 pg/mL) gave a more regulated EMT response over 7 days, while a higher dose (200 pg/mL) was used to induce a more rapid EMT response in the lens explants over this same time period. To determine the impact of FGF-2 on TGF-β2-induced lens EMT, we cotreated TGF-β2-treated LECs with either a low proliferating dose of recombinant human FGF-2 (5 ng/mL: 233-FB, R&D systems) or a high fibre-differentiating dose of FGF-2 (200 ng/mL) [10,50]. Control explants had no growth factors added to the media.

### 2.3. Assessment of Cell Morphology and Immunofluorescence

Cultured LEC explants were monitored and photographed daily over 7 days using phase contrast microscopy (Leica FireCam imaging, Leica Microsystems, Version 1.5, 2007). To examine the extent of how transdifferentiated cells modulated the underlying lens capsule, some treated explants were rinsed with filtered Milli-Q H_2_O to debride all cells from the explant to completely expose the underlying lens capsule. Phase contrast images were captured before and after rinsing.

Following the different growth factor treatments, at set time points, the explants were fixed in 10% neutral buffered formalin (NBF; HT501320-9.5L, Sigma-Aldrich Corp) for 10 min, followed by 3 × 5 min rinses in phosphate-buffered saline (PBS) supplemented with BSA (0.1%, *v*/*w*; PBS/BSA). The cells were permeabilized using PBS/BSA supplemented with Tween-20 (0.05%, *v*/*v*; 3 × 5 min), followed by subsequent rinses in PBS/BSA (2 × 5 min). The explants were then incubated at room temperature for 30 min with 3% normal goat serum (NGS diluted in PBS/BSA, *w*/*v*), before adding the primary antibodies; anti-mouse α-SMA (A2547, Sigma-Aldrich Corp.), anti-alpha tropomyosin (Tpm; α/9d; provided by Prof. Gunning, University of New South Wales, Sydney, NSW, Australia), anti-rabbit β-catenin (ab6302, Abcam, Fremont, CA, USA), anti-β-crystallin, and anti-total-Smad2/3 (t-Smad2/3: 8685, Cell Signaling Tech., Danvers, MA, USA), all diluted 1:100 in NGS/PBS/BSA. The explants were incubated overnight at 4 °C, followed by rinsing in PBS/BSA (3 × 5 min). The respective secondary antibodies were then applied for a 2 h incubation at room temperature: goat anti-rabbit IgG Alexa-Fluor^®^ 488 (ab150077, Abcam), and goat anti-mouse Alexa-Fluor^®^ 594 (ab150116, Abcam), all diluted 1:1000 in PBS/BSA. Three 5 min rinses in PBS/BSA were followed before a 5 min application of 3 μg/mL bisbenzimide (H33342 trihydrochloride, Hoechst counterstain, B2261, Sigma-Aldrich) diluted in PBS/BSA. The explants were rinsed again before mounting with 10% glycerol in PBS and imaged using epifluorescence microscopy (Leica DMLB 100S with DFC-450C camera, Leica Application Suite, Version 4.8, 2021).

### 2.4. SDS-Page and Western Blotting

Cultured lens epithelial explants at set time points were rinsed in cold PBS. The central and peripheral regions of the explants were isolated using a scalpel blade to delineate each region. A central square of tissue, no more than a third of the explant diameter (central LECs, CLECs), and the remaining surrounding peripheral LECs (PLECs) were isolated separately. CLEC and PLEC protein was harvested, pooled into allocated Eppendorf tubes, and lysed with cold radioimmunoprecipitation assay (RIPA) lysis buffer containing 150 mM NaCl, 0.5% sodium deoxycholate, 0.1% Sodium dodecyl sulphate (SDS), 1 mM sodium orthovanadate, 1 mM NaF, 50 mM Tris-HCl (pH 7.5), 0.1% Triton X-100, phosphatase (PhosSTOP^TM^), and protease (cOmplete^TM^) EASYpacks inhibitor tablets (04906837001 and 05892970001; Roche Applied Science, Basel, Switzerland). LECs were homogenized and centrifuged for 10 min at 4 °C (14,400× *g*) for lysate/supernatant separation. Quantification of the total lens protein of each supernatant sample was conducted using a Pierce^TM^ Micro bicinchoninic acid (BCA) protein assay reagent kit (23235; Thermo Fisher Scientific).

LEC protein sample lysates were prepared using 5% 2-mercaptoethanol (M6250, Sigma-Aldrich) combined with 2× Laemmli sample buffer at a 1:1 (*v/v*) ratio (1610737, Bio-Rad Laboratories, NSW, Australia). For electrophoresis, 10 μg of protein lysates were loaded onto 12% SDS-PAGE gels for 20 min at 70 V followed by 2 h at 120 V. LEC protein was then transferred onto an immobilon^®^-PSQ polyvinylidene fluoride (PVDF) membrane (ISEQ00010, Merck Millipore, Rahway, NJ, USA) for 1 h at 100 V. PVDF membranes were incubated in 2.5% BSA blocking buffer diluted in tris-buffered saline with 0.1% Tween-20 (TBST) and incubated for 1 h with agitation at room temperature. Primary antibodies were added to the membranes and left overnight to incubate (at 4 °C): anti-mouse α-SMA, anti-GAPDH (G8795, Sigma-Aldrich Corp.), anti-tropomyosin alpha, and anti-β-crystallin, t-Smad2/3, phospho-Smad2/3 (p-Smad2/3, 8828, Cell Signalling Tech., Danvers, MA, USA), phospho-ERK1/2 (p-ERK1/2, 9101, Cell Signalling Tech.), and total-ERK1/2 (t-ERK1/2, 9102, Cell Signalling Tech.), all diluted in blocking buffer/TBST at 1:1000, apart from α-SMA and GAPDH (1:2000). Following overnight incubation, the membranes were rinsed in TBST (3 × 5 min) and incubated with the appropriate horseradish peroxidase (HRP)-conjugated secondary antibodies for 2 h at room temperature: HRP-conjugated goat anti-rabbit IgG (7074, Cell Signalling Tech.) and horse anti-mouse IgG (7076, Cell Signalling Tech.), diluted in TBST at 1:5000. The membranes were rinsed in TBST (3 × 10 min) followed by the application of an immobilon chemiluminescent HRP substrate for 3–5 min (WBKLS0500, Merck Millipore). Protein chemiluminescent signals were imaged using Bio-Rad ChemiDoc^TM^ MP imaging.

Following immunolabeling, PVDF membranes were stripped for 10 min in stripping buffer (10% SDS, 0.5 M Tris HCl pH 6.8, Milli-Q H_2_O, and 0.8% 2-mercaptoethanol) with gentle agitation. The membranes were then washed in TBST (3 × 5 min) and re-blocked in blocking buffer/TBST for 1 h. Following blocking, the membranes were probed for loading control GAPDH (1:2000, 1 h) and incubated with an HRP-conjugated horse anti-mouse secondary antibody for 1 h prior to chemiluminescent signalling analysis. Protein densitometry was carried out using Bio-Rad ImageLab software (Version 6.1.0, 2019).

### 2.5. Statistical Analysis

For each experimental analysis, three independent experiments were carried out. For every experiment, a minimum of three individual replicates (*n* = 3) per treatment group (different treatment of explants) were used. For Western blotting, each group contained up to eight explants to isolate central and peripheral lens cells that were randomly obtained from different P21 rats. For measuring changes in protein expression, we used densitometry to calculate the selected protein intensity relative to the loading control (GAPDH).

For Western blot experiments examining differences in Smad-dependent (Smad2/3) and Smad-independent (MAPK/ERK1/2) activity, phosphorylated protein expression was calculated using the following ratio: relative phosphorylated density per total protein.

Prior to the use of one-way analysis of variance (ANOVA), several assumptions were tested and confirmed; we assumed equal standard deviation (SD) and residuals appeared normally distributed. Based on these confirmed assumptions, we compared the differences among the means of all treatment groups using one-way ANOVA, followed by Tukey’s multiple comparisons post-hoc test. All data acquired were plotted appropriately using GraphPad Prism software version 9.0 (GraphPad Software Inc., San Diego, CA, USA).

To quantify the spatial differences in t-Smad2/3 immunoreactivity, six separate images of central and peripheral regions were captured per explant across three randomized explants per treatment group, over three individual experiments. Nuclear and cytoplasmic localisation of t-Smad2/3 was manually counted using ImageJ’s Cell Counter plugin. The mean percentage of t-Smad2/3 nuclear and cytoplasmic fluorescence was calculated and statistically analysed using GraphPad Prism.

Tabled data were represented as the standard error of the mean (±SEM) and probability values, where *p* < 0.05 was considered statistically significant.

## 3. Results

### 3.1. FGF-2 Promotes a Spatially Dependent TGF-β2-Induced EMT Response in Lens Epithelial Explants

We examined the efficacy of different doses of FGF-2 in modulating the effect of TGF-β2 on lens epithelial cells induced to undergo EMT. Using phase contrast microscopy, control LECs without FGF-2 or TGF-β2 treatment (Figure 1A,E), as well as explants treated with only a low dose of FGF-2, demonstrated no significant morphological changes and retained their epithelial phenotype over the culture period. When the lens epithelial explants were treated with a low dose of TGF-β2 (50 pg/mL), this promoted an EMT response across the entire explant, similar to a higher dose (200 pg/mL) of TGF-β2, albeit at a slower rate, consistent with previous studies [42].

With different dose combinations of FGF-2 and TGF-β2, most cells in the explants underwent a uniform EMT response over 5 days, with the exception of cells in the explants cotreated with a relatively high dose of FGF-2 (200 ng/mL) and the lower dose of TGF-β2 (Figure 1), where we observed a differential response between CLECs and PLECs (Table 1, Appendix A).

The cells in the lens epithelial explants treated with the high a fibre-differentiating dose of FGF-2 elongated over 5 days (Figure 1B,F), compared to the control LEC explants (no growth factor treatment; Figure 1A,E). This FGF-induced fibre differentiation response was more pronounced in PLECs compared to CLECs (Figure 1B,F). LECs in explants treated with a low dose of TGF-β2 (Figure 1C,G) displayed prominent EMT by day 5, with the LECs losing their uniform packing and adhesion as they transdifferentiated into myofibroblastic cells. TGF-β2 treatment also led to increased cellular blebbing (refractile bodies) and apoptotic cell loss, evident by areas of bare lens capsule that displayed prominent signs of capsular modification in the form of wrinkling throughout the explant. When the explants were cotreated with TGF-β2 and FGF-2, CLECs underwent similar morphological transformations by day 5 (Figure 1D) to that seen with TGF-β2-treatment alone (Figure 1C,G). In contrast, PLECs in the explants cotreated with FGF-2/TGF-β2 showed no evidence of EMT (Figure 1H), instead demonstrating morphological changes more consistent with that observed in the explants treated with FGF-2 alone (Figure 1B,F).

PLECs in cotreated explants exhibited changes in the LEC phenotype as early as day 3 (Appendix A). Debridement of all cells at this time revealed the underlying lens capsule with no apparent capsular modulation (no folds or wrinkles) in either control (Appendix A) or FGF-2 treated (Appendix A) explants. In the TGF-β2-treated explants, increased capsular modulation was apparent with wrinkling and folds in the central explant regions (Appendix A) and in the peripheral regions (Appendix A). Consistent with the differential cell response in the central and peripheral regions of the F/t-cotreated explants (Figure 2A,A1,A3), the explants exhibited capsular modulation only in the central explant region (Figure 2A2), with no capsular wrinkling observed in the peripheral region (Figure 2A4).

We observed that with ongoing culture (up to 7 days), regardless of the explant region or treatment, all of the cells exposed to TGF-β2 (200 pg/mL) are lost by 7 days (Appendix A); however, in the cotreated explants (F/t), with continual supplementation of the media with FGF-2 (200 ng/mL) after day 3 of culture, this promoted cell survival, whereby we continue to observe many myofibroblastic cells in the central region of the explants (Appendix A) and, similarly, relatively normal lens cells at the periphery of the explants are also maintained (Appendix A).

### 3.2. FGF-2 Promotes Spatial Differences in Labelling for EMT and Fibre Differentiation Markers in TGF-β2-Treated LECs

#### 3.2.1. Immunofluorescent Labelling

We used immunofluorescence to characterise the different cell types in explants treated with TGF-β2 and/or FGF-2 over 3 days, labeling for lens fibre differentiation markers, β-crystallin, and alpha-tropomyosin (α/9d), as well as the EMT marker, α-SMA (Figure 3). Isotype controls for all of the antibodies show little to no specific labelling. Control LECs throughout the explant exhibited no reactivity for β-crystallin and/or α-SMA after 3 days of culture (Figure 3A), labelling only for α/9d (Figure 3B). FGF-2-treated LECs exhibited strong reactivity for β-crystallin throughout the explant (Figure 3C,I), with stronger labelling in PLECs (Figure 3I). Treatment with FGF-2 did not promote α-SMA reactivity in any cultured lens epithelia. FGF-2-treated CLECs presented diffuse α/9d-reactivity (Figure 3F), while PLECs had a more defined reactivity for α/9d, highlighting actin filaments in the elongating, differentiating fibre cells (Figure 3L). LECs treated with only TGF-β2 displayed clear evidence of an EMT response, with strong reactivity for α-SMA, with no β-crystallin observed throughout the explant (Figure 3D,J).

TGF-β2-treated CLECs had a highly specific localisation of α/9d to actin stress fibres (Figure 3G), which were also very prominent in PLECs (Figure 3M). Unlike cells treated with only FGF-2 or only TGF-β2, that had a relatively uniform label for the different markers across the entire explant, in the FGF-2/TGF-β2 cotreated explants, we observed distinct spatial differences in the labelling of the markers, consistent with our earlier morphological observations. The CLECs in the TGF-β2/FGF-2-treated explants predominantly labelled for α-SMA with little to no β-crystallin reactivity at day 3 (Figure 3E), similar to the explants treated with only TGF-β2 (Figure 3D,J). In contrast, the PLECs in these same cotreated explants displayed the inverse label, with strong reactivity primarily for β-crystallin in elongated cells, with few neighboring smaller cells immunolabelling for α-SMA (Figure 3K). This differential β-crystallin and α-SMA reactivity in the cotreated explants was sustained up to 5 days of culture (Appendix A). Stronger labelling for α/9d was also observed throughout the cotreated explants (Figure 3H,N), highlighting the marked elongation of peripheral fibre-like cells (Figure 3K,N), as well as central myofibroblastic cells (Figure 3E,H).

#### 3.2.2. Western Blotting

*Alpha-Smooth Muscle Actin*. We quantified protein changes in the treated explants using Western blotting. CLECs had a significant increase in α-SMA when treated with TGF-β2, compared to the relatively lower levels in the control (NT) and FGF-2-treated explants (*p* < 0.0001) (Figure 4A,C). FGF-2 treatment did not impact α-SMA levels in the CLECs compared to the control cells (*p* = 0.3429). In the FGF-2/TGF-β2 cotreated explants, there was a significant reduction in α-SMA levels in the CLECs relative to the TGF-β2 alone CLECs (*p* < 0.0001). In fact, these CLECs in the cotreated explants displayed no significant difference in levels of α-SMA compared to the CLECs of the control (*p* > 0.9999) or FGF-2 alone (*p* = 0.3249) explants. In the PLECs of the FGF-2/TGF-β2 cotreated explants, consistent with the reduced EMT response, there were reduced α-SMA levels when compared to the CLECs, comparable to the lower α-SMA levels seen in the PLECs of the control (*p* = 0.7371), FGF-2 (*p* > 0.9999)-, and TGF-β2-treated explants (*p* = 0.0053) (Figure 4B,D). The PLECs in the explants treated with FGF-2 alone did not have increased α-SMA levels when compared to the control cells (*p* = 7366); however, the PLECs in the explants treated with TGF-β2 alone had significantly increased α-SMA levels, compared to the control (*p* = 0.0203) and the FGF-2-treated (*p* = 0.0053) explants.

*β-crystallin*. When compared to the control cells, there was no significant difference in the levels of β-crystallin in the CLECs of the explants treated with FGF-2 (*p* = 0.8742) (Figure 4A,C). Treatment with TGF-β2 did not significantly increase levels of β-crystallin in the CLECs compared to the control (*p* = 0.5844), FGF-2 (*p* = 0.2260) or the cotreated explants (*p* = 0.1459). We did observe a significant decrease in β-crystallin in the CLECs of the cotreated explants, relative to the control (*p* = 0.0160) and FGF-2 (*p* = 0.0044)-treated explants (Figure 4A,C). Treatment of the explants with FGF-2 significantly increased β-crystallin levels in the PLECs when compared to the PLECs of the control (*p* = 0.0374) and the TGF-β2-treated explants (*p* = 0.0003) (Figure 4B,D). The PLECs in the explants treated with TGF-β2 alone had reduced β-crystallin levels when compared to the control PLECs (*p* = 0.0222). The PLECs of the explants cotreated with FGF-2/TGF-β2 had slightly elevated β-crystallin levels in comparison to the PLECs of the control (*p* = 0.6396) or the TGF-β2-treated (*p* = 0.0056) explants (Figure 4B,D).

Alpha-Tropomyosin. α/9d levels were significantly elevated only in the CLECs and PLECs of the TGF-β2-treated explants, when compared to the corresponding cells of all other treatment groups (Figure 4A–D). For the CLECs, levels of α/9d in the control explants were reduced in both the FGF-2 (*p* = 0.0924)- and FGF-2/TGF-β2-treated explants (*p* = 0.0959) and were significantly reduced when compared to the elevated α/9d levels found in the CLECs of the TGF-β2-treated explants (control vs. TGF-β2, *p* = 0.0034; FGF-2 vs. TGF-β2, *p* = 0.0002) (Figure 4C). In the PLECs, there was no obvious difference in the levels of α/9d across all of the treatment groups (Figure 4B), except for elevated levels in the PLECs of the TGF-β2-treated explants as mentioned (control vs. TGF-β2, *p* = 0.0103; FGF-2 vs. TGF-β2, *p* = 0.0039; TGF-β2 vs. FGF-2/TGF-β2, *p* = 0.0249) (Figure 4D).

### 3.3. Impact of FGF-2 on TGF-β2-Mediated Intracellular Signalling

#### 3.3.1. Nuclear Translocation of Smad2/3

Given that FGF-2 can differentially regulate TGF-β2-mediated LEC behaviour, we tested its impact on TGF-β2 mediated Smad2/3-signalling. Active TGF-β2-signalling is evident with the nuclear translocation of phosphorylated Smad2/3 (Figure 5).

After 2 h of culture, in both the control (Figure 5A,E) and the FGF-2 (Figure 5B,F)-treated explants, we do not see any Smad2/3 nuclear localisation: 0.45–2.81% nuclear labelling (Table 2, Figure 5I,J). In contrast, distinct nuclear localisation of Smad2/3 was evident throughout the TGF-β2-treated explants (Figure 5C,G): 86.44–88.8% nuclear labelling. In the lens epithelial explants cotreated with FGF-2/TGF-β2, we observed prominent nuclear translocation of Smad2/3 in the CLECs (Figure 5D,I): 44.87% nuclear labelling; however, in the PLECs the Smad2/3-labelling was primarily cytosolic with significantly reduced nuclear labelling: 19.85% (Table 2, Figure 5H,J).

#### 3.3.2. Smad2/3-Signalling

Treatment of the explants with FGF-2 did not impact p-Smad2/3 levels in CLECs when compared to similar levels in the control CLECs (*p* = 0.9768, Figure 6A,B) or the PLECs (*p* = 0.9310, Figure 6D,E) after 6 h of culture. Consistent with our immunofluorescent nuclear localisation of Smad2/3 (Figure 5), TGF-β2 significantly elevated p-Smad2/3 levels in the CLECs compared to the CLECs of the control explants (*p* = 0.0061) and the FGF-2-treated explants (*p* = 0.0101) (Figure 6A,B). In the CLECs of the explants cotreated with FGF/TGF-β2, there was no significant difference in p-Smad2/3 levels when compared to the CLECs of the TGF-β2 (*p* = 0.5944)- and the FGF-2-treated explants (*p* = 0.0585); however, p-Smad2/3 levels significantly increased in the cotreated CLECs compared to the control explants (*p* = 0.0334) (Figure 6A,B). In the PLECs, the TGF-β2 treated explants exhibited elevated p-Smad2/3 levels in comparison to the control (*p* = 0.0178), FGF-2 alone (*p* = 0.0403), and cotreated PLEC explants (*p* = 0.6496, Figure 5D,E) (Figure 6D,E). Compared to the control- and FGF-2-treated PLEC explants, cotreatment with FGF-2/TGF-β2 increased p-Smad2/3 levels (*p* = 0.0933 for the control, *p* = 0.2122 for FGF-2) (Figure 6D,E).

#### 3.3.3. MAPK/ERK1/2-Signalling

Levels of phosphorylated ERK1/2 (p-ERK1/2) remained constant in the CLECs of the control and FGF-2-treated (*p* = 0.7703, Figure 6A,C) explants after 6 h but were elevated in the PLECs of the FGF-2-treated explants, compared to the control PLECs (*p* = 0.0140, Figure 6D,F). TGF-β2 treatment of the explants slightly increased p-ERK1/2 activity in the CLECs compared to the levels in the CLECs of the control (*p* = 0.0227) and FGF-2 treated explants (*p* = 0.0880) (Figure 6A,C). In contrast, the PLECs of the TGF-β2-treated explants had decreased p-ERK1/2 levels compared to the PLECs of the control (*p* = 0.6711) and FGF-2-treated explants (*p* = 0.0033) (Figure 6D,F). The CLECs in the explants cotreated with FGF-2/TGF-β2 had reduced p-ERK1/2 levels in comparison to the CLECs in the TGF-β2-treated (*p* = 0.0174), FGF-2-treated (*p* = 0.6641), and control explants (*p* = 0.9972) (Figure 6A,C). Interestingly, the PLECs of the cotreated explants demonstrated a significant increase in their p-ERK1/2 levels in contrast to the low levels in the PLECs of the control (*p* = 0.0195) and TGF- β2 treated explants (*p* = 0.0242) (Figure 6D,F).

## 4. Discussion

The present study has demonstrated the impact of FGF-2 on the behaviour of lens epithelial cells induced to undergo EMT in response to TGF-β. A lens-fibre-differentiating dose of FGF-2 was able to block TGF-β2-induced lens EMT in only the peripheral LECs in explants (equivalent to the germinative region of the intact lens) and not in the central (more anterior) lens epithelia. As seen in previous wholemount rat lens epithelial cell explant models, we have demonstrated that CLECs and PLECs exposed to TGF-β2 alone undergo an EMT response, with no evidence of lens fibre differentiation [24,41,51]; however, in combination with FGF-2, FGF-2 potentiates this TGF-β2-induced activity, with elevation of canonical Smad2/3 signalling activity, as well as EMT-associated markers, more so in the CLECs.

For our lens epithelial explant model, we used relatively low doses of TGF-β2 (50 and 200 pg/mL) to induce an EMT response across a short culture period [48,52]. This dose is physiologically representative of concentrations of TGF-β2 in its mature (approx. 100 pg/mL) and total (>3000 pg/mL) forms observed in situ [53]. In addition, it is comparable to active forms of TGF-β2 (approx. 100–400 pg/mL) found in cataractous patients [53,54,55,56,57]. Our use of a lower TGF-β2 dose contrasts to other studies that have used much higher doses (0.5–1.5 ng/mL) to elicit an EMT response in rodent lens cells [13,14,16], which could potentially lead to off-target growth factor signalling activity. Exogenous addition of FGF-2 at a high dose encourages all lens epithelial cells (both CLECs and PLECs) to undergo a change in cell morphology typical of fibre differentiation [3,4,58,59]. In explants cotreated with TGF-β2 and FGF-2, FGF-2 appeared to protect PLECs from TGF-β-induced EMT by promoting a fibre differentiation response. We showed that the PLECs in these FGF-2/TGF-β2 cotreated explants had prominent elongated fibres, reminiscent of many earlier studies from our laboratory [59]. An elevated dose of TGF-β2, was able to prevent any fibre differentiation in the PLECs, leading to an enhanced EMT response in both central and peripheral cells. In addition, we demonstrated that the PLECs in the cotreated explants did not exhibit contractile properties as evidenced by the lack of capsular wrinkling in this region, unlike the region of the CLECs undergoing EMT. The inhibition of lens epithelial cell contraction by FGF despite the presence of TGF-β has been shown in other fibrotic models to be dose dependent, such as in bovine LECs cultured in collagen I gel [60] and valvular interstitial cells (VICs) modelling valvular fibrosis [61], which is also correlated with reduced α-SMA expression.

We not only report how TGF-β can impact FGF-induced lens cell responsiveness but how FGF in turn influences TGF-β-induced responses in LECs, the main focus of our study. When we examined for changes in cytoskeletal and stress-fibre associated proteins, the CLECs in TGF-β/FGF-cotreated lens explants exhibited predominant α-SMA localisation and little to no β-crystallin, suggesting that these cells cannot resist the EMT process, despite the presence of a high differentiating dose of FGF-2. The co-influence of FGF-2 and TGF-β on fibre differentiation, epithelial, and EMT-associated marker expression has previously been reported in other models, including human lung epithelial cells and rat alveolar epithelial-like cells [62], as well as E10 chick lens epithelial cells [13], and the lenses of postnatal mice [10]. Despite the CLECs in the TGF-β/FGF-cotreated explants undergoing a prominent EMT response, we noted reduced α-SMA and α/9d levels, compared to the TGF-β-alone-treated explants, suggesting that FGF-2 may be potentially compromising Tpm activity (a recruiter for actin assembly) and attenuating α-SMA stress fibre association. It has been previously shown that FGF may influence Tpm activity and expression, as well as cell biomechanics, in the presence of TGF-β [14]. For example, when murine LECs (MLECs) are cotreated with FGF-2/TGF-β2, the loss of Tpm1 corresponded with decreased α-SMA reactivity [14]. This same study also confirmed FGF-2 modulation of Tpm in HLECs, when cotreated with TGF-β2, with a significant reduction in both Tpm1 and Tpm2 levels [14]. We localized Tpm (α/9d) in LECs undergoing different phenotypic changes in fibre differentiation, but more compellingly in cells undergoing EMT, where it was associated with the α-SMA-reactive stress fibres of myofibroblasts. This may be attributed to the fact that the α/9d antibody we used specifically targets several isoform splice variant products of the αTm gene (TPM1), including Tpm1.4, Tpm1.6–1.9, and Tpm2.1, with some cross-reactivity also for Tpm3.1 [63]. Tpm1.6, Tpm2.1, and Tpm3.1 have all previously been characterized as being stress-fibre associated and are suggested to play a role in TGF-β induced EMT [64,65,66]. FGF has shown a role in propagating stress-induced EMT in conjunction with TGF-β in other pathologies, such as wound healing in mice skin keratinocytes [45] and in the tumor stromal cell microenvironment of prostate fibroblasts [47]. Consistent with our findings, Koike et al. (2020) [45] found that FGF-2 could not solely induce EMT in mice keratinocytes; however, in keratinocytes cotreated with FGF-2 and TGF-β1, there was a significant upregulation of cell migratory/motility and EMT-associated genes (e.g., *VIM* and *SNAI2*), similar to keratinocytes with only TGF-β1 stimulation. In a non-transformed mouse mammary gland epithelial cell line (NMuMG), TGF-β modulated FGF receptor activation, increased FGF-2 cell sensitivity, and promoted an EMT response through activation of ERK1/2 signalling [49], highlighting the synergistic signalling role of these two growth factors.

To determine how FGF-2 was modulating and antagonizing TGF-β2-induced EMT in PLECs of cotreated LEC explants, we explored changes in their signalling activity, namely changes to Smad2/3 and ERK1/2. In cotreated lens epithelial explants, we saw stronger signalling for the respective pathways in different regions; CLECs undergoing EMT had more pronounced p-Smad2/3 activity, while PLECs undergoing fiber differentiation had more pronounced p-ERK1/2-signalling. FGF is a well-known regulator of ERK1/2 within the lens, with its marked phosphorylation evident in lens cells within minutes post treatment [3,4,67,68]. While ERK1/2 has been shown to be required for lens epithelial cell proliferation, it is also very important for lens fiber differentiation [4,67,68,69,70]. This differs from TGF-β2-induced EMT, where we found that while ERK is also involved in this EMT process, blocking ERK1/2 does not completely block TGF-β2-mediated EMT progression in lens epithelia [39,48,52]. In fact, canonical Smad2/3-signalling is most evident in EMT, as shown here in our cotreated CLECs, and in many earlier studies examining TGF-β2-induced lens EMT [11,13,38,52,71,72].

While we and others have shown FGF-2 is not able to promote Smad2/3-signalling in LECs [10], FGF-2 was shown to impede nuclear localisation of Smad2/3 in PLECs in explants cotreated with TGF-β2; however, in CLECs of these same explants, FGF-2 appeared to have less of an impact on TGF-β2-induced Smad2/3-activity. How FGF-2 directly blocks Smad2/3 activity in PLECs is not clear but given the strong ERK1/2 activation in these cells, this may favour lens fibre differentiation and cell survival, as we see here and has been shown by others [2,3,67,73]. Conversely, FGF-mediated ERK1/2 signalling can correlate with the upregulation of TGF-β activity, as seen in other fibrosis models [14,49,74,75,76], as well as the current study where TGF-β-induced CLECs are associated with elevated ERK1/2-signalling. Similar to the current study, in valvular interstitial cells (VICs) modelling valvular fibrosis, it was shown that inhibition of this fibrosis was dependent on FGF-2-mediated MAPK signalling when cotreated with TGF-β1 [61]. This study demonstrated that FGF (10 ng/mL) prevented Smad3 nuclear localisation in VICs cotreated with TGF-β1 (5 ng/mL), and at higher doses (100 ng/mL), it was able to perturb TGF-β1-mediated α-SMA expression [61], highlighting the ability of FGF to modulate canonical TGF-β signalling activity and downstream gene expression.

Although not completely understood, crosstalk between FGF and TGF-β signalling has proven influential in mediating various fibrotic disorders and carcinoma progression. For example, a study implementing mouse tumor-associated endothelial cells (TECs) demonstrated how FGF can promote a differential cell response by reducing TGF-β-induced contractile and myofibroblastic properties, while concurrently promoting cell proliferation and motility [75]. A similar finding was observed in primary human dermal fibroblasts (HDFs), whereby FGF-2 with TGF-β1 cotreatment, both positively and negatively regulated fibroblast transition into cancer-associated fibroblasts (CAFs) [77]. This same study also showed how this FGF-2/TGF-β1 treatment of HDFs can downregulate common CAF-activated and EMT-associated markers (e.g., *ACTA2*, *ITGA11*, and *COL1A1*) as well as upregulating cell motility and morphogenetic genes (e.g., *HGF* and *BMP2*) [77].

Research into the mechanisms surrounding differential types of PCO involving lens fibre cell types is ongoing and it is believed to be due to FGF/TGF-β interactions during EMT induction [13,15,16,38,74,78]. As FGF is a major factor influencing normal lens fibre differentiation, it is important to understand what promotes aberrant fibre differentiation during pearl PCO development at the lens equator [13,24,35,79]. In situ, for ASC and for post-operative PCO, more anterior lens epithelial cells are likely exposed to a high insult of TGF-β, and relatively low levels of FGF are normally found in the aqueous humour. At the lens equator, however, epithelial cells in the posterior chamber are regularly exposed to elevated levels of FGF, and regardless of any increased TGF-β levels, the cells here likely undergo aberrant fibre differentiation, leading to pearl PCO. This may result from the heightened sensitivity to FGF of these peripheral LECs, namely due to their elevated levels of high-affinity FGF receptor tyrosine kinase (RTK) receptors, compared to the central lens epithelia [6,58,80,81]. In situ, during lens fibrosis, we do not see EMT resulting in cell death, likely due to survival growth factors present within the ocular media. Given the findings from the current study, we propose that FGF is a putative survival factor in situ, maintaining fiber cells at the lens equator and the myofibroblastic phenotype leading to fibrotic PCO. Further studies investigating differences/changes in levels of FGF and TGF-β receptors, between central and peripheral lens cells in cotreated explants, may be a key factor in determining lens cell fates in situ. We also cannot rule out that changes in the expression of RTK antagonists, such as Sprouty and Spreds [69,70,82], including those more specific for FGF, such as Sef [83], in these active regions of the lens may be protective of peripheral LECs from any aberrant TGF-β insult of which they have previously been reported to block [69].

## 5. Conclusions

A fine balance between levels of FGF-2 and TGF-β2 can promote differential responses in lens epithelial cells. More specifically, this responsiveness is spatially regulated, with the anterior central lens epithelia more sensitive to EMT induction, whilst peripheral cells primarily undergo fibre differentiation in the presence of high levels of FGF, avoiding apoptotic cell death associated with EMT. This induced heterogeneous population of cells in lens epithelial explants may provide an alternative model better suited to the study of the cellular processes at play in situ, leading to the formation of ASC, and more importantly both fibrotic and pearl forms of PCO.

## Figures and Tables

**Figure 1 cells-12-00827-f001:**
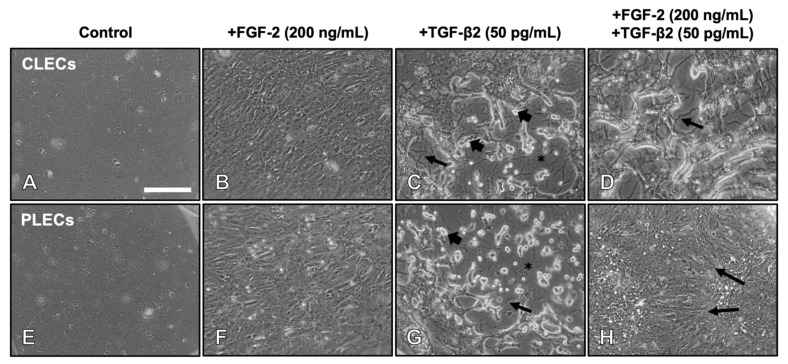
FGF-2 promotes a spatially dependent TGF-β2-induced EMT response in lens epithelial explants. Control LECs maintained a cobblestone-like epithelial phenotype after 5 days (**A**,**E**). FGF-2-induced cell elongation typical of lens fibre differentiation (**B**,**F**). TGF-β2 induced EMT, highlighted by elongated myofibroblastic cells, with prominent cell blebbing/refractile bodies ((**C**,**G**), arrowheads) and loss of cells exposing the lens capsule (asterisk) with capsular wrinkling (arrows). FGF-2 and TGF-β2 cotreated explants led to EMT of CLECs (**D**) but a fibre differentiation response in PLECs ((**H**), arrows). Scale bar: 200 μm.

**Figure 2 cells-12-00827-f002:**
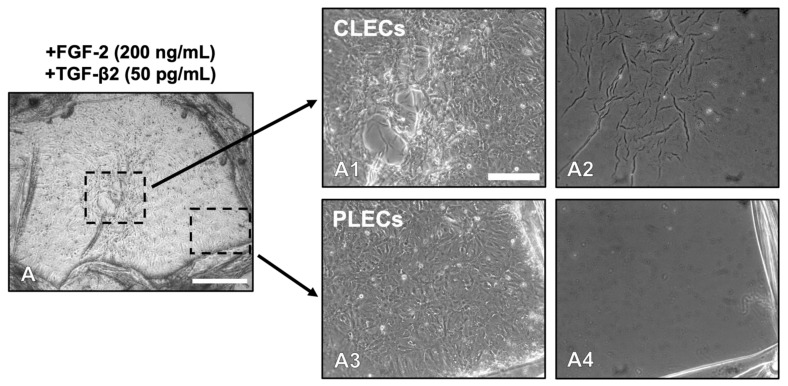
Lens capsule modulation during TGF-β2-induced EMT. Seventy-two h post treatment, the explants were rinsed consecutively in filtered Milli-Q H_2_O to remove all lens epithelial cells and view the underlying lens capsule. In the F/t cotreated explants (**A**,**A1**,**A3**), after cell removal (**A2**,**A4**), capsular wrinkling was only apparent in regions that were previously populated with CLECs (**A2**), with no wrinkling visible in regions that were previously populated with PLECs (**A4**). Scale bar: 400 µm (**A**), 200 µm (**A1**–**A4**).

**Figure 3 cells-12-00827-f003:**
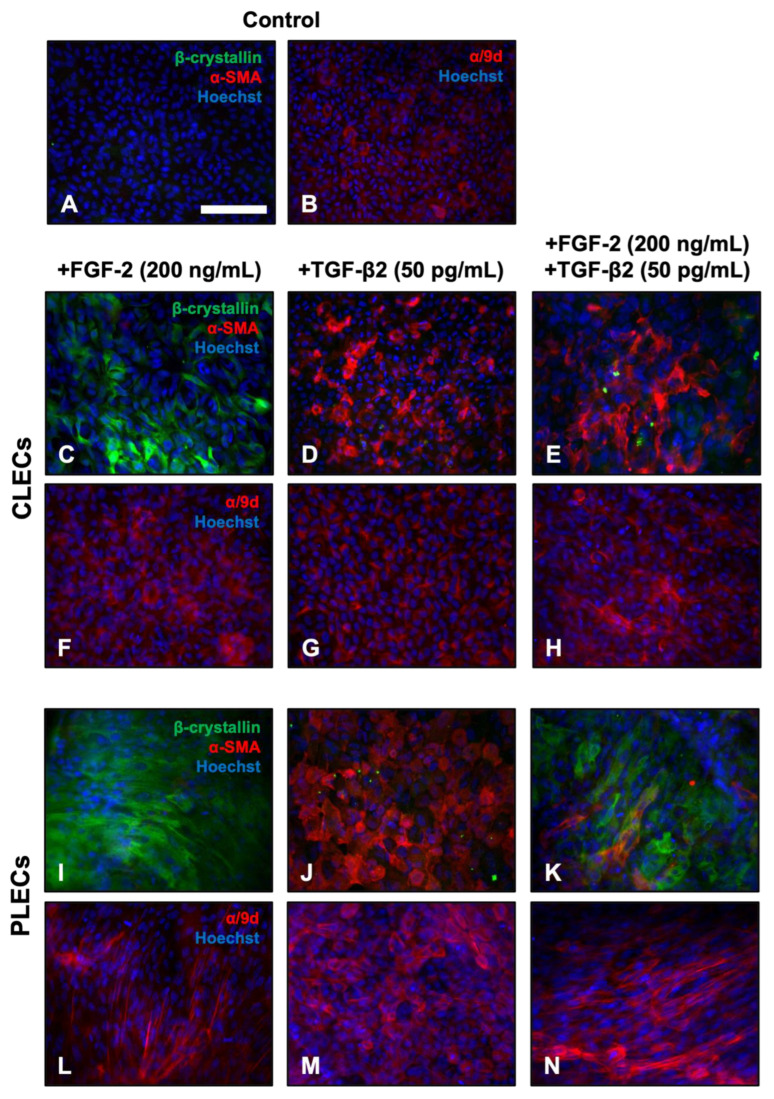
FGF-2 modulates EMT and fibre differentiation markers in TGF-b2-treated LECs. Immunolabeling of β-crystallin (green), α-SMA (red), and alpha-tropomyosin (α/9d; red), counterstained with Hoechst nuclear stain (blue), in CLECs (**C**–**H**) and PLECs (**I**–**N**) following 3 days of culture with no growth factors (Control, (**A**,**B**)), FGF-2 (200 ng/mL; (**C**,**F**,**I**,**L**)), and TGF-β2 (50 pg/mL; (**D**,**G**,**J**,**M**)), or cotreated with FGF-2 and TGF-β2 (**E**,**H**,**K**,**N**). Images are representative of three independent experiments. Scale bar: 100 μm.

**Figure 4 cells-12-00827-f004:**
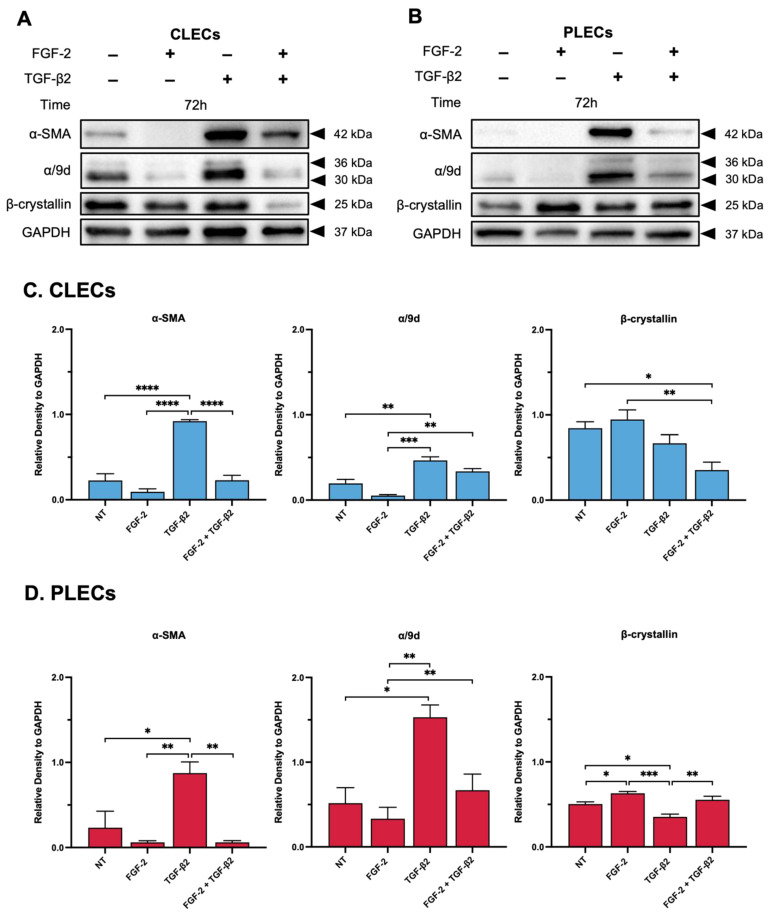
FGF-2 modulates levels of different protein markers in TGF-β2-treated LECs. Representative Western blot for alpha-tropomyosin (α/9d), α-SMA, and β-crystallin in the control (non-treated, NT), FGF-2, TGF-β2, and FGF-2/TGF-β2 cotreated CLECs (**A**,**C**) and PLECs (**B**,**D**). Protein levels were normalized relative to GAPDH levels (**C**,**D**). One-way ANOVA with the mean ± SEM and post-hoc Tukey’s multiple comparisons test (* *p* < 0.0332, ** *p* < 0.0021, *** *p* < 0.002, **** *p* < 0.001).

**Figure 5 cells-12-00827-f005:**
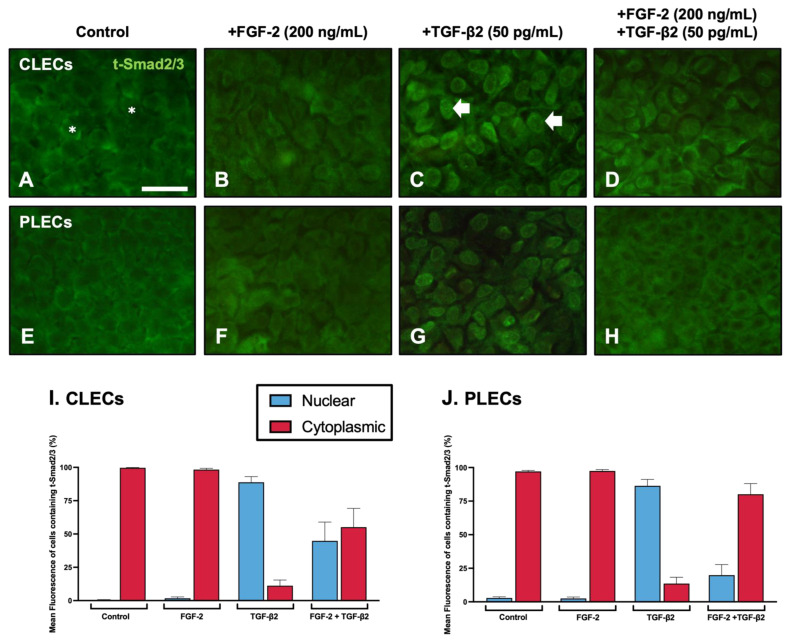
FGF-2 modulates TGFβ-2-induced Smad2/3 nuclear translocation. Immunolabelling of total Smad2/3 (t-Smad2/3, green) in LEC explants following two hours of culture (**A**–**H**). CLECs (**A**–**D**) and PLECs (**E**–**H**) in explants treated with no growth factors (control; (**A**,**E**)), FGF-2 (200 ng/mL; (**B**,**F**)), TGF-β2 (50 pg/mL; (**C**,**G**)), or cotreated with FGF-2/TGF-β2 (**D**,**H**). Examples of cytosolic (asterisks) and nuclear (arrowheads) localisation. Mean percentage (±SEM) fluorescence of cells with nuclear (blue) and cytoplasmic (red) t-Smad2/3 localisation in CLECs (**I**) and PLECs (**J**). Scale bar: 50 μm.

**Figure 6 cells-12-00827-f006:**
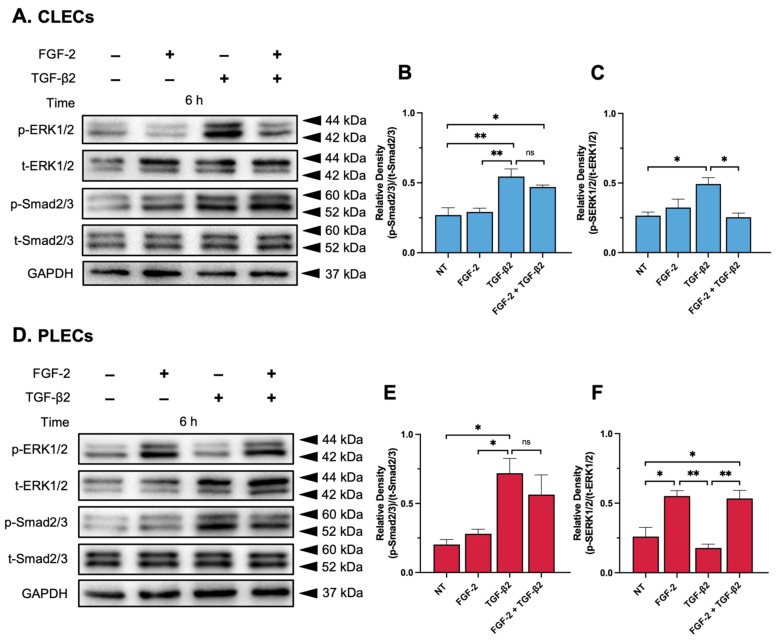
FGF-2 modulates TGF-β2-signalling. Representative Western blots demonstrating protein levels of phosphorylated and total Smads (p-Smad2/3 and t-Smad2/3) and MAPK/ERK1/2 (p-ERK1/2 and t-ERK1/2) in the control (non-treated, NT), FGF-2, TGF-β2, and FGF-2/TGF-β2 cotreated CLECs (**A**–**C**) and PLECs (**D**–**F**). The protein densitometry analysis shows changes in the levels of relative phosphorylation of Smad2/3 (**B**,**E**) and ERK1/2 (**C**,**F**). One-way ANOVA with the mean ± SEM and post-hoc Tukey’s multiple comparisons test (ns = not significant, * *p* < 0.0332, ** *p* < 0.0021).

**Table 1 cells-12-00827-t001:** Regional, dose-dependent effects of FGF-2 and TGF-β2 on LEC explants.

Treatment	Concentration		Regional Cell Response in Explant	
	FGF-2 (ng/mL)	TGF-β2 (pg/mL)	FGF-2 (ng/mL)	TGF-β2 (pg/mL)
f/t	5	50	EMT	EMT
f/T	5	200	EMT	EMT
F/t	200	50	EMT	*Fibre Differentiation*
F/T	200	200	EMT	EMT

Low dose FGF-2 (f); low dose TGF-β2 (t); high dose FGF-2 (F); high dose TGF-β2 (T).

**Table 2 cells-12-00827-t002:** Nuclear vs. cytoplasmic localisation of t-Smad2/3 in CLECs and PLECs.

Localisation of t-Smad2/3 (Total Mean % of Fluorescence)				
Treatment	CLECs		PLECs	
	Nuclear	Cytoplasmic	Nuclear	Cytoplasmic
Control	0.35 ± 0.351	99.65 ± 0.351	2.91 ± 0.833	97.09 ± 0.833
FGF-2 (200 ng/mL)	1.70 ± 0.988	98.30 ± 0.988	2.55 ± 1.047	97.45 ± 1.047
TGF-β2 (50 pg/mL)	88.80 ± 4.217	11.20 ± 4.217	86.40 ± 5.444	13.60 ± 5.444
FGF-2 (200 ng/mL) + TGF-β2 (50 pg/mL)	44.87 ± 14.058	55.13 ± 14.058	19.85 ± 9.090	80.15 ± 9.090

The values are the mean percentage of fluorescence of t-Smad2/3 reactivity (%) ± SEM. *Abbreviations*: Central lens epithelial cells (CLECs); control (non-treated explants); peripheral lens epithelial cells (PLECs). Refer to Figure 5.

## Data Availability

Not applicable.

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
