# Peer review of "FGF-2 Differentially Regulates Lens Epithelial Cell Behaviour during TGF-β-Induced EMT"

_cells, 2023, doi:10.3390/cells12060827_

Round 1
Reviewer 1 Report
Authors aimed to demonstrate that high dose of FGF could be protective to EMT caused by TGFb treatment in cells originating from rat lens explants. Presented manuscript is composed properly and clearly describes all conducted experiments. The described results and drown conclusions are supported by the presented data. However, I have noticed a few issues that authors could improve in this work. Please find my comments below.
1. Authors obtained the lens epithelial cells from rat lens explants. The methodology of gathering the cells is understandable, however it could be specified how the explants were pinned to the culture dishes (line 128).
2. The good practice in in vitro studies is the confirmation of obtained cell origin by determining at least its specific molecular markers by immunostaining or their expression assessment at mRNA level. No data on the harvested cell identification/confirmation was presented.
3. I suggest moving sentences describing the aim of this study to a new paragraph at the end of introduction to expose them (line 102 or 105).
4. All used reagents have the description of its catalogue number and producer. The description could be added also to the plasticware like culture dishes.
5. Authors did not describe how the dose of TGFb was selected. The dose selection was explained in the discussion, however it could be mentioned in material and methods to justify its selection.
6. The procedure of western blotting is well described, however it is confusing how many replications were applied for WBs. I suggest adding new subsection describing each experiment design highlighting the group sizes. Also, it would be informative to provide descriptive statistics for each examination.
7. The title “2.5. Statistical Analysis” should be moved to next page.
8. The statistical analysis is briefly described. Authors mentioned that there were at least 3 replicates per group, but it is not clear when (for which experiment). Moreover, authors claim that they used ANOVA with post hoc Tukey to evaluate the statistical differences between groups, but do the collected data met the assumptions of this test? Authors should describe how they tested the data prior applying the ANOVA.
9. The figure 1 control group appears to have no cells. The scale bar seems to be inappropriate, by the comparison of cell size to this bar it looks like the cells are quite small and it is hard to assume what is their size. The scale bars should be presented all the images, authors could also add the information about the magnification.
10. The isotype controls in figure 3 and 5 should be presented.
11. It would be beneficial to this report adding the data on gene expression analysis. The presented methods are based only on proteomics.
Author Response
We thank the reviewer for their evaluation and comments, and we have addressed these accordingly. As needed, we have revised the manuscript, with all emendations in manuscript marked in red.
REVIEWER 1
Authors aimed to demonstrate that high dose of FGF could be protective to EMT caused by TGFb treatment in cells originating from rat lens explants. Presented manuscript is composed properly and clearly describes all conducted experiments. The described results and drown conclusions are supported by the presented data. However, I have noticed a few issues that authors could improve in this work. Please find my comments below.
- Authors obtained the lens epithelial cells from rat lens explants. The methodology of gathering the cells is understandable, however it could be specified how the explants were pinned to the culture dishes (line 128).
While we do not literally pin the explants to the dish with a pin, by pressing down on the explant with fine forceps, they remain attached to the base of the culture dish. This method has been in place for over 4 decades and is well published. We have expanded our methods to specify the explant pinning process.
- The good practice in in vitro studies is the confirmation of obtained cell origin by determining at least its specific molecular markers by immunostaining or their expression assessment at mRNA level. No data on the harvested cell identification/confirmation was presented.
The cells in this study were all primary lens epithelial cells collected fresh from weanling rats. They are all of known origin and there would be no need for further characterization to determine their identification/confirmation, as would be expected for passaged cell lines.
- I suggest moving sentences describing the aim of this study to a new paragraph at the end of introduction to expose them (line 102 or 105).
We have now created a new paragraph at the end of the Introduction to expose the aims more clearly.
- All used reagents have the description of its catalogue number and producer. The description could be added also to the plasticware like culture dishes.
We have now added more reference to the source of the materials used.
- Authors did not describe how the dose of TGFb was selected. The dose selection was explained in the discussion; however, it could be mentioned in material and methods to justify its selection.
The different doses we used for TGF-b have been well established in our model for over 30 years and are well documented. We trialed different doses for this study to establish the differential effects. We based our experiments on the doses we established in the primary experiments. We have now added a comment on this in the Methods as requested.
- The procedure of western blotting is well described; however it is confusing how many replications were applied for WBs. I suggest adding new subsection describing each experiment design highlighting the group sizes. Also, it would be informative to provide descriptive statistics for each examination.
Yes, they meet the assumptions of the test and we have now better clarified the number of replicates used for our experiments in the Methods, as well as the statistics applied.
- Tissue samples were randomised and pooled from different animals.
- We assumed Guassian distribution and equal standard deviations, leading to residuals passing the normality test.
- ANOVA: we compared the mean of each column with the mean of every other column (treatment groups).
- The title “2.5. Statistical Analysis” should be moved to next page.
This has been automatically adjusted.
- The statistical analysis is briefly described. Authors mentioned that there were at least 3 replicates per group, but it is not clear when (for which experiment). Moreover, authors claim that they used ANOVA with post hoc Tukey to evaluate the statistical differences between groups, but do the collected data met the assumptions of this test? Authors should describe how they tested the data prior applying the ANOVA.
As we caried this out for all experiments, we more broadly stated that we used at least 3 replicates per group. We have better justified the rationale for using ANOVA with post hoc Tukey. Please refer to comment 6 earlier, and the revised Methods section.
- The figure 1 control group appears to have no cells. The scale bar seems to be inappropriate, by the comparison of cell size to this bar it looks like the cells are quite small and it is hard to assume what is their size. The scale bars should be presented all the images, authors could also add the information about the magnification.
The Figure 1 control group indeed has many cells. In fact, it has a complete sheet of cuboidal lens epithelia that are so tightly packed that it is hard to discern individual cells using phase contrast microscopy under lower magnification. This lack of cellular detail also emphasizes that these primary lens epithelia are tightly adherent to each other, and that this epithelial trait evidently changes once treated with TGFb or FGF. If you digitally zoom into the Figure, cell profiles are clearly evident. For comparison, Figure 2A4 indeed no longer contains cells, and its surface appears clearer.
As all images are taken at the same magnification, we have the one representative scale bar for the Figure. The scale bar measurement is already included in the Figure legend. We have used this style of scale bar and referencing for countless manuscripts without much issue in the past.
- The isotype controls in figure 3 and 5 should be presented.
The antibodies used in Figures 3 and 5 are commercially available and well characterized. They have been extensively used in many of our earlier studies and that of others, where isotype controls were included. We have now included a statement referencing our controls to this effect.
- It would be beneficial to this report adding the data on gene expression analysis. The presented methods are based only on proteomics.
This study is limited to protein analysis which is the transdifferentiation marker end-product. While we thank the reviewer for this suggestion, and appreciate that a transcriptome analysis will be informative, it is beyond the scope of this study, and we currently do not have the means to execute this effectively.
Reviewer 2 Report
A central tenant in the PCO field is that lens cells from the center of the anterior epithelium (CLECs) that survive cataract surgery tend to undergo EMT to myofibroblastic cells, whereas more peripheral lens epithelial cells (PLECs) are more susceptible to differentiation into lens fiber-like cells, with the latter process being largely responsible for the formation of Soemmering's ring. The basis for this clinically important difference in developmental fate is not, however, fully understood. In this manuscript, the authors subject explants of lens epithelial cells prepared from P21 rats to FGF and/or TGFB, two growth factors relevant to the etiology of PCO. As expected (and previously reported), culturing such explants in recombinant TGFB induces them to largely undergo EMT as assessed by expression of aSMA, whereas relatively high levels of FGF-2 causes them instead to upregulate the lens fiber cell marker B-crystallin. The interesting observation presented here is that cells from the central epithelium (CLECs) respond to a mixture of 200 ng/ml FGF-2 and 50 pg/ml TGFB mainly by turning on expression of aSMA, whereas epithelial cells from more peripheral parts of the explant (PLECs) preferentially upregulate B crystallin. This suggests that at least part of the reason that central and peripheral lens epithelial cells undergo different fates during the development of PCO could be related to the well-known differences in gene expression between these two populations of cells. Indeed, the authors propose that the greater expression of FGF receptors in peripheral vs central epithelial cells may account for the greater tendency of PLECs to differentiate into lens fiber cells. Given that 200 ng/ml FGF2 is a much stronger inducer of ERK activation in PLECs than 50 pg/ml TGFB (Fig 6D), this could also account for the author’s finding that PLECs have a higher apparent ratio of ERK-to-Smad2/3 signaling than CLECs.
The data are presented in a clear, convincing manner. There are however, some areas of the text that need clarification/correction.
1. Abstract: “Smad-dependent and independent signaling was increased in FGF-2/TGF-β2 co- treated CLECs, with heightened nuclear localization of Smad2/3, while PLECs featured more pronounced ERK1/2-signaling over Smad2/3 activation.”
“heightened” relative to what? I assume to similarly treated PLECs.
Pg 3: “We demonstrate that a high fiber differentiating dose of FGF is protective of TGF-β-induced EMT in peripheral lens epithelia; however, exacerbates the EMT response of central lens epithelia induced by TGF-β.” Fig 3 and 4 show that the level of aSMA is lower in TGFB/FGF co-treated CLECs than in CLECs treated with TGFB only.
Pg 9: “When compared to control cells, there was no significant difference in levels of β-crystallin in CLECs of explants treated with FGF-2 (P = 0.8742) (Figure 4A, 4C).” There is an obvious increase in β-crystallin staining in Fig 3C (FGF-treated CLECs) vs Fig 3A (intreated control explants), as would be expected based on previous studies in the rat explant system.
Pg 12: “Levels of phosphorylated ERK1/2 (p-ERK1/2) were elevated in CLECs of control …. explants, as well as in PLECs of control …. explants (P = 0.0140, Figure 6D, 6F) after 6 hours.” Elevated in controls relative to what?
Pg 14: “Compared to explants treated with FGF-2 alone, we saw an increased level of β-crystallin in PLECs co-treated with TGF-β.” Not in Fig 3 (3I vs 3K), or in Fig 4B and 4D. Omit this paragraph.
Pg 14: “CLECs in TGF-β/FGF-cotreated lens explants exhibited predominant α-SMA stress fiber localization. “ None of the images shown in Fig 3 show very obvious localization of aSMA in stress fibers.
Pg 15: “This differs to TGF-β2-induced EMT, where we found that while ERK is also involved in this process, it is not essential [39,48,52].
Given that one of these references is titled “ERK1/2-mediated EGFR-signaling is required for TGFβ-induced lens epithelial-mesenchymal transition,” and another “ERK1/2 signaling is required for the initiation but not progression of TGFβ-induced lens epithelial to mesenchymal transition (EMT),” I am not sure where this statement came from.
Pg 15: “however, in CLECs of these same explants, FGF-2 appeared to potentiate TGF-β2-induced Smad2/3-activity.” Not in Fig 5 (C vs D; I) or Fig 6 (A; B).
Pg 15: “where CLECs associated with elevated ERK1/2 signaling” Elevated relative to what?
Pg 16: “Based on these studies, it can be proposed that TGF-β-induced Smad-dependent and independent signaling is required for FGF expression during pro-fibrotic events.” Unless there is evidence for this in lens epithelial cells, I would omit this paragraph.
Pg 16: “The current study showed that despite weaker reactivity for the Smad2/3 complex in FGF/TGF-β treated PLECs, both CLECs and PLECs in cotreated explants favored stronger lower molecular weight immunolabeling for Smad3, suggesting FGF mediation of Smad3 activity.” Or the antibody could have higher affinity for pSmad3 than for pSmad2. Unless you account for this possibility, and demonstrate which band is Smad2 vs Smad3, and quantitate this finding, I would omit this speculation.
Supplemental Fig 2 has the same treatments (and same panel A) as is shown in Fig 1. What additional information is provided?
It would be nice to have a low-mag view of an entire 200 ng/ml FGF2 + 50 pg/ml TGFB co-treated explant so one could appreciate the relative distributions of aSMA and B crystallin.
Author Response
We thank the reviewer for their evaluation and comments, and we have addressed these accordingly. As needed, we have revised the manuscript, with all emendations in manuscript marked in red.
REVIEWER 2
- Abstract: “Smad-dependent and independent signaling was increased in FGF-2/TGF-β2 co- treated CLECs, with heightened nuclear localization of Smad2/3, while PLECs featured more pronounced ERK1/2-signaling over Smad2/3 activation.”
“heightened” relative to what? I assume to similarly treated PLECs.
Thank you for highlighting this, and we agree it is somewhat ambiguous. We have now reworded this statement as follows: “Smad-dependent and independent signaling was increased in FGF-2/TGF-β2 co-treated CLECs, that had a heightened number of cells with nuclear localization of Smad2/3 compared to PLECs that in contrast had more pronounced ERK1/2-signaling over Smad2/3 activation.”
- Pg 3: “We demonstrate that a high fiber differentiating dose of FGF is protective of TGF-β-induced EMT in peripheral lens epithelia; however, exacerbates the EMT response of central lens epithelia induced by TGF-β.” Fig 3 and 4 show that the level of aSMA is lower in TGFB/FGF co-treated CLECs than in CLECs treated with TGFB only.
Given this point is not validated by our data, we have rephrased this statement as follows.
“We demonstrate that a high fiber differentiating dose of FGF is protective of TGF-β-induced EMT in peripheral lens epithelia; however, this is not evident in central lens epithelia induced by TGF-β.”
- Pg 9: “When compared to control cells, there was no significant difference in levels of β-crystallin in CLECs of explants treated with FGF-2 (P = 0.8742) (Figure 4A, 4C).” There is an obvious increase in β-crystallin staining in Fig 3C (FGF-treated CLECs) vs Fig 3A (intreated control explants), as would be expected based on previous studies in the rat explant system.
We welcome the reviewers point and offer an explanation. Previous studies from our laboratory have clearly shown that FGF indeed promotes increased β-crystallin staining in LECs. In contrast to these earlier studies, what we examine here is β-crystallin staining in response to FGF after only 3 days of culture, and also in P21 rats. Previous studies show robust labeling for β-crystallin after 5 days culture and in much younger rat LECs. Earlier studies also show that the responsiveness of LECs to FGF decreases with age (Lovicu and McAvoy, 1992), as evident in the current study. Here we have used an earlier time point to primarily show the onset of both aSMA and β-crystallin staining, before the increased cell death associated with EMT compromises our ability to do so, especially via WB. While Figure 3C IF label may not correlate directly with the WB for β-crystallin staining data in Figure 4A/4C, we believe that we have yet to reach a significant level of β-crystallin using this more sensitive assay. Clearly Figure 3C shows some cells that are accumulating β-crystallin, but just not at sufficient levels at this early stage of culture.
- Pg 12: “Levels of phosphorylated ERK1/2 (p-ERK1/2) were elevated in CLECs of control …. explants, as well as in PLECs of control …. explants (P = 0.0140, Figure 6D, 6F) after 6 hours.” Elevated in controls relative to what?
Thank you for highlighting this oversight on our part. We have now corrected it to read as follows: “Levels of phosphorylated ERK1/2 (p-ERK1/2) remained constant in CLECs of control and FGF-2-treated (P = 0.7703, Figure 6A, 6C) explants after 6 hours, but were elevated in PLECs of FGF-2-treated explants compared to control PLECs (P = 0.0140, Figure 6D, 6F).”
- Pg 14: “Compared to explants treated with FGF-2 alone, we saw an increased level of β-crystallin in PLECs co-treated with TGF-β.” Not in Fig 3 (3I vs 3K), or in Fig 4B and 4D. Omit this paragraph.
Given this Discussion point is clearly not validated by our data, we have deleted this paragraph as suggested.
- Pg 14: “CLECs in TGF-β/FGF-cotreated lens explants exhibited predominant α-SMA stress fiber localization.” None of the images shown in Fig 3 show very obvious localization of aSMA in stress fibers.
Given that the stress fibers are not readily discernible at the magnification we present here, and with epifluorescent microscopy, we concur with the reviewer and have not highlighted the localisation to stress fibers.
- Pg 15: “This differs to TGF-β2-induced EMT, where we found that while ERK is also involved in this process, it is not essential [39,48,52]. Given that one of these references is titled “ERK1/2-mediated EGFR-signaling is required for TGFβ-induced lens epithelial-mesenchymal transition,” and another “ERK1/2 signaling is required for the initiation but not progression of TGFβ-induced lens epithelial to mesenchymal transition (EMT),” I am not sure where this statement came from.
We have now clarified this statement based on the literature cited, to state: “This differs to TGF-β2-induced EMT, where we found that while ERK is also involved in this EMT process, blocking ERK1/2 does not completely block TGF-β2 mediated EMT progression in lens epithelia [39,48,52]”.
- Pg 15: “however, in CLECs of these same explants, FGF-2 appeared to potentiate TGF-β2-induced Smad2/3-activity.” Not in Fig 5 (C vs D; I) or Fig 6 (A; B).
Given this Discussion point is clearly not validated by our data, we have now reworded reference to this as follows; “however, in CLECs of these same explants, FGF-2 appeared to have less of an impact on TGF-β2-induced Smad2/3-activity.
- Pg 15: “where CLECs associated with elevated ERK1/2 signaling.” Elevated relative to what?
We have now clarified this to read: “where TGF-β-induced CLECs associated with elevated ERK1/2-signaling.”
- Pg 16: “Based on these studies, it can be proposed that TGF-β-induced Smad-dependent and independent signaling is required for FGF expression during pro-fibrotic events.” Unless there is evidence for this in lens epithelial cells, I would omit this paragraph.
While we have some room to speculate, it does merit further work in lens, and as a result we opted to delete this paragraph as suggested.
- Pg 16: “The current study showed that despite weaker reactivity for the Smad2/3 complex in FGF/TGF-β treated PLECs, both CLECs and PLECs in cotreated explants favored stronger lower molecular weight immunolabeling for Smad3, suggesting FGF mediation of Smad3 activity.” Or the antibody could have higher affinity for pSmad3 than for pSmad2. Unless you account for this possibility, and demonstrate which band is Smad2 vs Smad3, and quantitate this finding, I would omit this speculation.
Again, given the limited data we have to justify this speculation, we have opted to delete this paragraph as suggested.
- Supplemental Fig 2 has the same treatments (and same panel A) as is shown in Fig 1. What additional information is provided?
Fig 1 and Suppl Fig 2a were originally the one Figure. In the process of splitting them, we inadvertently used the same image for part of both. We have now corrected this for Suppl Fig 2A.
The main difference between Figure 1 and Suppl Fig 2 is the length of culture period. Figure 1 clearly shows the phenomenon we are characterising after 5 days of culture, while Suppl Figure 2 shows this same albeit weaker response after 3 days when it was first evident; this is also the time point that we use for all our immunolabeling, as it contains more cells compared to Day 5 where many cells undergoing EMT are lost. Based on this, we respectively request to retain this supplementary image for those wanting to correlate the immunofluorescence with the equivalent stage phase contrast images.
- It would be nice to have a low-mag view of an entire 200 ng/ml FGF2 + 50 pg/ml TGFB co-treated explant so one could appreciate the relative distributions of aSMA and B-crystallin.
While we agree that this option would be ideal, as you can appreciate, low power epifluorescence does not demonstrate the same intensity of label than a higher power image would. The only way to overcome this would be to take many higher power images and make a montage that still may not present true given the fact that some images may bleach after continual exposure.
Round 2
Reviewer 1 Report
The manuscript has been improved. In am satisfied with authors answers.